# Cycle Class Consistency with Distributional Optimal Transport and Knowledge Distillation for Unsupervised Domain Adaptation

**Tuan Nguyen**[1]     **Van Nguyen**[2]     **Trung Le**[1]     **He Zhao**[1]     **Quan Hung Tran**[3]     **Dinh Phung**[1,4]

[1]Department of Data Science and AI, Monash University, Australia
[2]The University of Adelaide, Australia
[3]Adobe Research, San Jose, CA, USA
[4]VinAI Research, Vietnam

## Abstract

Unsupervised domain adaptation (UDA) aims to transfer knowledge from a model trained on a labeled source domain to an unlabeled target domain. To this end, we propose in this paper a novel cycle class-consistent model based on optimal transport (OT) and knowledge distillation. The model consists of two agents, a teacher and a student cooperatively working in a cycle process under the guidance of the distributional optimal transport and distillation manner. The OT distance is designed to bridge the gap between the distribution of the target data and a distribution over the source class-conditional distributions. The optimal probability matrix then provides pseudo labels to learn a teacher that achieves a good classification performance on the target domain. Knowledge distillation is performed in the next step in which the teacher distills and transfers its knowledge to the student. And finally, the student produces its prediction for the optimal transport step. This process forms a closed cycle in which the teacher and student networks are simultaneously trained to conduct transfer learning from the source to the target domain. Extensive experiments show that our proposed method outperforms existing methods, especially the class-aware and OT-based ones on benchmark datasets including Office-31, Office-Home, and ImageCLEF-DA.

## 1 INTRODUCTION

Unsupervised domain adaptation (UDA) allows us to transfer knowledge from a model trained on a source domain with labels to a target domain without any labels. To cope with structural data more efficiently and effectively, deep domain adaptation (DDA) [Ganin and Lempitsky, 2015]

has been proposed and widely studied [Nguyen et al., 2019, 2020, Phung et al., 2021]. To tackle the data shift issue and learn domain-invariant features, DDA aims to bridge the distribution gap between the source and target domains in a latent space using a feature extractor. Guided by this principle, most of the existing works in DDA propose minimizing a divergence between the source and target distributions in the latent space. Popular choices of divergence include the Jensen-Shannon (JS) divergence [Ganin and Lempitsky, 2015, Tzeng et al., 2015, Shu et al., 2018], the maximum mean discrepancy (MMD) distance [Gretton et al., 2007, Long et al., 2015], and the Wasserstein (WS) distance [Shen et al., 2018, Lee et al., 2019, Le et al., 2021a].

Recently, Optimal transport (OT) [Villani, 2008, Santambrogio, 2015], a powerful tool in mathematics with rich and rigorous theories, has been widely applied in deep domain adaptation [Courty et al., 2017b,a, Damodaran et al., 2018, Redko et al., 2019, Lee et al., 2019, Xie et al., 2019, Xu et al., 2020, Nguyen et al., 2021a,b, Le et al., 2021b, Nguyen et al., 2021c]. From the conceptual perspective, OT-based methods encourage the target samples to move towards the source samples by minimizing a transportation cost. However, since the transportation cost usually engages the pairs of target and source samples without considering label information of the source samples, the movement of the target samples to the source domain seems to be unaware of the class regions in that domain, hence cannot resolve the label shift issue [Tachet des Combes et al., 2020]. Although OT has been initially used for solving this problem [Courty et al., 2017b, Damodaran et al., 2018], the performance of the existing methods is still less satisfactory compared with the state-of-the-art ones.

In this paper, we propose a novel distributional OT that enables the incorporation of the source label information when engaging and matching target and source samples. Specifically, in the source domain we consider that one label is associated with a conditional distribution over all the samples conditioned on that label. Next, we define a distribution over these conditional distributions of all the

*Accepted for the 38th Conference on Uncertainty in Artificial Intelligence* (UAI 2022).

labels in the source domain. In the target domain where there are no labels, we also consider a distribution over all the target samples. With the two distributions for the source and target domains respectively, we formulate the DA problem as the computation of the OT distance between the two distributions. The OT transport plan gives us the information of how a target sample related to the source samples by taking into account the source domain labels. The challenge here is how to define the cost function, which indicates the transport cost of OT between a target sample and a source class-conditional distribution. To tackle this challenge, we propose a cycle class consistency framework in which we leverage the advantages of knowledge distillation (KD) which has recently obtained outstanding achievements [Tian et al., 2020, Zhao et al., 2020, Tejankar et al., 2021, Feng et al., 2021]. We name our proposed approach *Cycle Class COnsistency with Optimal Transport and Knowledge Distillation for Unsupervised Domain Adaptation* (COOK). In summary, our contributions in this paper include:

- We propose a novel distributional OT which seeks the optimal matching between the target and source examples taking into account the source label information for reducing the label and data shift, two challenging problems of UDA.

- We connect KD and OT to further improve the performance of class-aware UDA methods via proposing a cycle class consistency framework where the teacher and student networks cooperatively work in a distillation process and support to reduce the mismatch between the target distribution and the source class-conditional distributions.

- We conduct experiments to compare our proposed COOK with the existing standard UDA methods, especially class-aware UDA methods (e.g., RADA [Wang et al., 2019b] and CAN [Kang et al., 2019]), and OT-based UDA methods (e.g., DeepJDOT [Damodaran et al., 2018], ETD [Li et al., 2020], and RWOT [Xu et al., 2020]). The experimental results show that our proposed method surpasses the baselines on the benchmark datasets including *Office-31*, *Office-Home*, and *ImageCLEF-DA*.

## 2   RELATED WORK

### 2.1   STANDARD DA

Deep domain adaptation has been intensively studied and shown appealing performance in various tasks and applications, notably in Ganin and Lempitsky [2015], Long et al. [2015], et al. [2017, 2018]. The core idea of DDA is to bridge the gap between source and target distributions in a joint space by minimizing a divergence between distributions induced from the source and target domains in

this space. Popular choices of divergence include Jensen-Shannon divergence [Ganin and Lempitsky, 2015, Tzeng et al., 2015, Shu et al., 2018]; maximum mean discrepancy distance [Gretton et al., 2007, Long et al., 2015]; and WS distance [Shen et al., 2018, Lee et al., 2019, Le et al., 2021a]. Some recent works have exploited different aspects of UDA for improving the performance [Kurmi et al., 2019, Wang et al., 2019a, Chen et al., 2019, Hu et al., 2020]. Typically, CADA [Kurmi et al., 2019] considered the probabilistic certainty estimate of various regions and used these certainty estimate weights for improving the classifier performance on the target dataset. GSDA [Hu et al., 2020] introduced a novel method named Hierarchical Gradient Synchronization to model the synchronization relationship among the local distribution pieces and global distribution, aiming for more precise domain-invariant features.

### 2.2   OPTIMAL TRANSPORT BASED DA

Optimal transport theory has been applied to domain adaptation in Courty et al. [2017b,a], Damodaran et al. [2018], Redko et al. [2019], Lee et al. [2019], Xie et al. [2019], Xu et al. [2020]. Particularly, Lee et al. [2019] proposed using sliced Wasserstein distance for domain adaption, whereas Xie et al. [2019] proposed SPOT in which the optimal transport plan is approximated by a pushforward of a reference distribution, and cast the optimal transport problem into a minimax problem. Recent OT-based DA work (RWOT) [Xu et al., 2020] leveraged spatial prototypical information and intra-domain structures of image data to reduce the negative transfer caused by target samples near decision boundaries. Moreover, Courty et al. [2017b] proposed an idea to connect the theory of optimal transport and domain adaptation, which later inspired an OT-based deep DA method (DeepJDOT) [Damodaran et al., 2018]. Another recent work (ETD) [Li et al., 2020] tackled the bottlenecks of OT in UDA by developing an attention-aware OT distance to measure the domain discrepancy under the guidance of the prediction-feedback. Our proposed approach is totally different from existing OT based DA approaches in which we examine an OT distance discrete distribution over source class-conditional distributions and the target data distribution. By investigating this specific OT distance and minimizing it, we can guide target examples moving to an appropriate source class on the latent space for mitigating both data and label shifts.

### 2.3   CLASS-AWARE DA

Some recent approaches [Wang et al., 2019b, Kang et al., 2019] leverage the useful information from the label space to improve the quality of the alignment between the source and target domains. Wang et al. [2019b] proposed a novel relationship-aware adversarial domain adaptation (RADA) algorithm. It first uses a single multi-class domain discrimi-

nator to enforce the learning of inter-class dependency structure during domain-adversarial training. After that, it aligns this structure with the inter-class dependencies that are characterized from training the label predictor on source domain. Furthermore, the authors imposed a regularization term in order to penalize the structure discrepancy between the inter-class dependencies estimated from domain discriminator and label predictor. With this alignment, RADA makes the adversarial domain adaptation aware of the class relationships. Kang et al. [2019] proposed a contrastive adaptation network (CAN) which optimizes a new metric modeling the intra-class domain discrepancy and the inter-class domain discrepancy. In particular, the authors introduced a new contrastive domain discrepancy (CDD) objective to enable class-aware UDA. CAN aims to faciliate the optimization with CDD (established on maximum mean discrepancy (MMD) [Long et al., 2015]).

# 3 BACKGROUND

In what follows, we present the background of OT for two discrete distributions, which is used in our work. Consider two discrete distributions: $\mathbb{P}^1 = \sum_{i=1}^M \pi_i^1 \delta_{\mathbf{x}_i^1}$ and $\mathbb{P}^2 = \sum_{j=1}^N \pi_j^2 \delta_{\mathbf{x}_j^2}$ where $\boldsymbol{\pi}^1 = \left[\pi_i^1\right]_{i=1}^M$ and $\boldsymbol{\pi}^2 = \left[\pi_j^2\right]_{j=1}^N$ are probability masses, $\left\{\mathbf{x}_i^1\right\}_{i=1}^M$ and $\left\{\mathbf{x}_j^2\right\}_{j=1}^N$ are the sets of atoms, and $\delta_{\mathbf{x}}$ is the Dirac delta distribution concentrated at the atom $\mathbf{x}$. Let $c\left(\mathbf{x}_i^1, \mathbf{x}_j^2\right)$ be a cost function. The OT distance between $\mathbb{P}^1$ and $\mathbb{P}^2$ w.r.t. the cost function $c$ is defined as

$$\min_{A \in \mathbb{R}_+^{M \times N}} \sum_{i=1}^M \sum_{j=1}^N a_{ij} c\left(\mathbf{x}_i^1, \mathbf{x}_j^2\right), \qquad (1)$$

where $A = [a_{ij}] \in \mathbb{R}_+^{M \times N}$ of non-negative elements satisfying $\sum_{j=1}^N a_{ij} = \pi_i^1, \forall i \in \{1, ..., M\}$ and $\sum_{i=1}^M a_{ij} = \pi_j^2, \forall j \in \{1, ..., N\}$.

In addition, $a_{ij} \in [0; 1]$ is interpreted as the probability to match $\mathbf{x}_i^1$ and $\mathbf{x}_j^2$ or to transport $\mathbf{x}_i^1$ to $\mathbf{x}_j^2$, which suffers the cost $c\left(\mathbf{x}_i^1, \mathbf{x}_j^2\right)$. Therefore, the sum $\sum_{i=1}^M \sum_{j=1}^N a_{ij} c\left(\mathbf{x}_i^1, \mathbf{x}_j^2\right)$ can be viewed as the total cost to match $\mathbb{P}^1$ and $\mathbb{P}^2$ or to transport $\mathbb{P}^1$ to $\mathbb{P}^2$. By solving the optimization problem in Eq. (1), we aim to find the optimal transportation matrix $A^*$ which minimizes the total cost.

# 4 DISTRIBUTIONAL OPTIMAL TRANSPORT APPROACH FOR CLASS-AWARE UDA

## 4.1 PROBLEM FORMULATION

We consider the standard setting of unsupervised domain adaptation in which we have a labeled dataset $\mathbb{D}^S =$

$\left\{\left(\mathbf{x}_i^S, y_i^S\right)\right\}_{i=1}^{N_S}$ from a source domain and an unlabeled dataset $\mathbb{D}^T = \left\{\mathbf{x}_i^T\right\}_{i=1}^{N_T}$ from a target domain. We assume that data examples $\mathbf{x}_i^S, \mathbf{x}_i^T \in \mathbb{R}^d$ and the categorical labels $y_i^S \in \{1, 2, ..., M\}$ where $M$ is the number of classes. For the sake of notion simplification, we overload $\mathbb{D}^S$ and $\mathbb{D}^T$ to represent the empirical joint distributions of the source and target domains. We denote $\mathbb{P}^S$ and $\mathbb{P}^T$ as the data distributions of the source and target domains respectively. Moreover, given a class $m$, we further denote $\mathbb{P}_m^S$ as the $m$-th class-conditional distribution of the source domain (i.e., the distribution with the density function $p^S\left(\mathbf{x} \mid y = m\right)$).

## 4.2 MOTIVATION

For our proposed approach, we consider an OT distance of two discrete distributions. The first one is the discrete distribution whose atoms are the target examples $\mathbf{x}^T$ (i.e., $\mathbf{x}_i^1 = \mathbf{x}_i^T$ in Eq. (1)), while the second one is the discrete distribution whose atoms are the source class-conditional distributions $\mathbb{P}_m^S$ (i.e., $\mathbf{x}_j^2 = \mathbb{P}_m^S$ in Eq. (1)). The cost $c\left(\mathbf{x}_i^T, \mathbb{P}_m^S\right)$ is defined as the negative log likelihood $-\log p_m^S\left(\mathbf{x}_i^T\right) = -\log p^S\left(\mathbf{x}_i^T \mid y = m\right)$. Hence, if a target sample $\mathbf{x}_i^T$ is more likely to be a sample from $\mathbb{P}_m^S$, the log likelihood $\log p_m^S\left(\mathbf{x}_i^T\right)$ is higher, meaning that the cost $c\left(\mathbf{x}_i^T, \mathbb{P}_m^S\right) = -\log p_m^S\left(\mathbf{x}_i^T\right)$ becomes smaller. As shown in Figure 1, by examining the OT distance between two aforementioned distributions, we aim to find the best match between a given target sample $\mathbf{x}_i^T$ and a source class-conditional distribution $\mathbb{P}_m^S$.

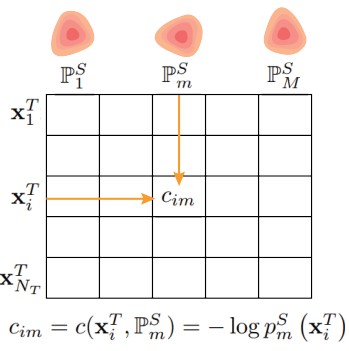

$$c_{im} = c(\mathbf{x}_i^T, \mathbb{P}_m^S) = -\log p_m^S\left(\mathbf{x}_i^T\right)$$

Figure 1: We consider the OT distance between two distributions: the first one has atoms as the target examples $\mathbf{x}^T$ and the second one has atoms as the class-conditional distributions $\mathbb{P}_m^S$. The cost function $c(\mathbf{x}_i^T, \mathbb{P}_m^S) = -\log p_m^S\left(\mathbf{x}_i^T\right) = -\log p^S\left(\mathbf{x}_i^T \mid y = m\right)$.

## 4.3 DISTRIBUTIONAL OPTIMAL TRANSPORT

We define $\mathcal{P}^S = \sum_{m=1}^M \pi_m \delta_{\mathbb{P}_m^S}$, where $\delta$ is the Dirac delta distribution and the mixing proportion $\boldsymbol{\pi} \in \Delta_M := \left\{\boldsymbol{\alpha} \in \mathbb{R}^M : \boldsymbol{\alpha} \geq \mathbf{0} \text{ and } \|\boldsymbol{\alpha}\|_1 = 1\right\}$ with the number of

classes $M$. Obviously, $\mathcal{P}^S$ is a discrete distribution of distributions wherein $\mathcal{P}^S$ takes $\mathbb{P}_m^S$ with the probability $\pi_m$. As mentioned in the motivation section, we now examine an OT distance between $\mathbb{P}^T$ and $\mathcal{P}^S$, we aim at matching target examples to the source class-conditional distributions in which a target example is absolutely guided to match the source class-conditional distribution corresponding to its ground-truth label.

In the sequel, we inspect an OT distance between $\mathbb{P}^T$ and $\mathcal{P}^S$ in which we define the cost $c\left(\mathbf{x}_i, \mathbb{P}_m^S\right)$ to match a target sample $\mathbf{x}_i$ to $\mathbb{P}_m^S$ as $-\log p_m^S\left(\mathbf{x}_i\right)$. Let us denote $A = [a_{im}] \in \mathbb{R}^{N_T \times M}$ as the transportation matrix wherein $a_{im}$ represents the probability to match or transport $\mathbf{x}_i$ to $\mathbb{P}_m^S$. The OT distance between $\mathbb{P}^T$ and $\mathcal{P}^S$ w.r.t. the cost function $c$ and the mixing proportion $\boldsymbol{\pi}$ is defined as:

$$
\begin{aligned}
\mathcal{W}_{c,\boldsymbol{\pi}}\left(\mathbb{P}^T, \mathcal{P}^S\right) = \min_A \Bigg\{ &\sum_{i=1}^{N_T} \sum_{m=1}^{M} a_{im} c\left(\mathbf{x}_i, \mathbb{P}_m^S\right) : \\
&\sum_{m=1}^{M} a_{im} = \frac{1}{N_T}, \sum_{i=1}^{N_T} a_{im} = \pi_m \Bigg\}.
\end{aligned}
\tag{2}
$$

Similar to other DA works [Pan et al., 2008, Tzeng et al., 2015, Long et al., 2017], we employ a feature extractor $G$ to map both source and target examples to a latent space. We denote $\mathbb{Q}^S, \mathbb{Q}^T, \mathbb{Q}_m^S$, and $\mathcal{Q}^S$ as the corresponding distributions over the latent space induced by $\mathbb{P}^S, \mathbb{P}^T, \mathbb{P}_m^S$, and $\mathcal{P}^S$ via the feature extractor $G$. The OT distance in Eq. (2) is rewritten as:

$$
\begin{aligned}
\mathcal{W}_{c,\boldsymbol{\pi}}\left(\mathbb{Q}^T, \mathcal{Q}^S\right) = \min_A \Bigg\{ &\sum_{i=1}^{N_T} \sum_{m=1}^{M} a_{im} c\left(G\left(\mathbf{x}_i\right), \mathbb{Q}_m^S\right) : \\
&\sum_{m=1}^{M} a_{im} = \frac{1}{N_T}, \sum_{i=1}^{N_T} a_{im} = \pi_m \Bigg\}.
\end{aligned}
\tag{3}
$$

To encourage the target examples $G\left(\mathbf{x}_i\right)$ to move towards proper class regions of the source domain, we propose solving the following optimization problem (OP):

$$
\min_{G, \boldsymbol{\pi}} \mathcal{W}_{c,\boldsymbol{\pi}}\left(\mathbb{Q}^T, \mathcal{Q}^S\right).
\tag{4}
$$

With $c\left(G\left(\mathbf{x}_i\right), \mathbb{Q}_m^S\right) = -\log p_m^S\left(\mathbf{x}_i\right)$, minimizing the OT distance in Eq. (4) encourages the target example $G\left(\mathbf{x}_i\right)$ to move towards a $\mathbb{Q}_k^S \left(1 \leq k \leq M\right)$ with a high likelihood and $\mathbf{a}_i = [a_{im}]_m$ inspired to be close to the corresponding scaled one-hot vector $\frac{1}{N_T}\mathbf{1}_k$. Here we denote $\mathbf{1}_k$ as the one-hot vector with the $k$-th element being one.

## 5 CYCLE CLASS CONSISTENCY FRAMEWORK

### 5.1 COST FUNCTION AND KNOWLEDGE DISTILLATION

To define the cost function $c\left(G\left(\mathbf{x}_i\right), \mathbb{Q}_m^S\right)$ in Eq. (4), we build a classifier $h^S$ over the latent space, and rely on its output to compute the cost values. This classifier is first trained using the labeled source dataset $\mathbb{D}^S = \left\{\left(\mathbf{x}_i^S, y_i^S\right)\right\}_{i=1}^{N_S}$ by minimizing the empirical loss:

$$
\mathcal{L}^{src} = \frac{1}{N_S} \sum_{i=1}^{N_S} CE\left(\sigma\left(h^S\left(G\left(\mathbf{x}_i\right)\right)\right), y_i^S\right),
\tag{5}
$$

where $\sigma$ denotes a softmax function and $CE$ represents a cross-entropy loss. Recap that given a target example $\mathbf{x}_i$, $c\left(G\left(\mathbf{x}_i\right), \mathbb{Q}_m^S\right)$ captures the matching extent of $G\left(\mathbf{x}_i\right)$ and the class-conditional distribution $\mathbb{Q}_m^S$. Therefore, we can reasonably define $c\left(G\left(\mathbf{x}_i\right), \mathbb{Q}_m^S\right) = -\log \sigma_m\left(h^S\left(G\left(\mathbf{x}_i\right)\right)\right)$ (i.e., $\sigma_m\left(h^S\left(G\left(\mathbf{x}_i\right)\right)\right)$ is the predicted probability of $\mathbf{x}_i$ belonging to class $m$ by classifier $h^S$).

However, we find that $h^S$ is a well-trained classifier on the source domain, and can generalize poorly on the target domain due to the data and label shifts. Therefore, instead of using only one classifier trained to work well on both domains, we leverage knowledge distillation [Hinton et al., 2015, Tian et al., 2020, Tejankar et al., 2021] which includes the two-network architecture, a teacher $h^T$ and a student $h^S$. The teacher $h^T$ aims to be an expert on the target domain, while the student $h^S$, which classifies accurately on the source domain, is also able to generalize on the target domain via distilling knowledge from its teacher. When the generalization ability of $h^S$ is improved, the cost $c\left(G\left(\mathbf{x}_i\right), \mathbb{Q}_m^S\right)$ is computed more accurately to solve the OP in Eq. (3). Inspired by the work of Hinton et al. [2015], we perform knowledge distillation from the teacher $h^T$ to the student $h^S$ in the target domain by minimizing a distillation loss $\mathcal{L}^{dl}$ w.r.t. a temperature softmax function:

$$
\mathcal{L}^{dl} = \frac{1}{N_T} \sum_{i=1}^{N_T} CE\left(\sigma\left(\frac{h^S\left(G\left(\mathbf{x}_i\right)\right)}{\tau}\right), \sigma\left(\frac{h^T\left(G\left(\mathbf{x}_i\right)\right)}{\tau}\right)\right),
\tag{6}
$$

where $\tau$ is a temperature parameter. When setting $\tau > 1$, the teacher and student's predictions become softer, from which the student can capture "dark knowledge" [Hinton et al., 2015] from the teacher and effectively mimic the teacher's behaviour.

The student $h^S$ is now trained well in the source domain via Eq. (5), and is possible to generalize on the target domain via Eq. (6). To achieve this good generalization capability,

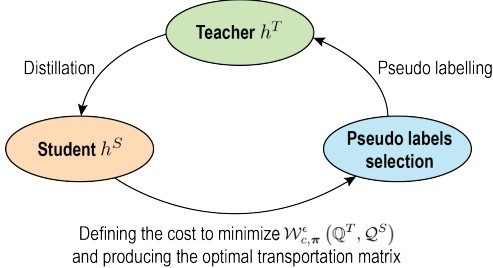

Figure 2: The proposed cycle class consistency framework.

we need to produce a teacher $h^T$ that is with good classification performance on the target domain. To this end, we propose minimizing a cross-entropy loss between the teacher's prediction and pseudo labels computed via the optimal transportation matrix $A^*$ after solving Eq. (3):

$$\mathcal{L}^{pl} = \frac{1}{n_T} \sum_{i=1}^{n_T} CE\left(\sigma\left(h^T\left(G\left(\mathbf{x}_i\right)\right)\right), \hat{y}_i^T\right), \quad (7)$$

where $\hat{y}^T$ are pseudo labels for unlabeled target samples. It is worth noting that only a subset of target samples with high-confidence pseudo labels is selected (i.e., $n_T < N_T$). In the next section, we discuss on how to compute these pseudo labels and our framework.

## 5.2 PSEUDO-LABEL SELECTION AND OUR FRAMEWORK

We now introduce the strategy to produce pseudo labels for unlabeled target samples. Let us return to the Eq. (3) where directly solving this OP is computationally expensive. Hence, we instead use an entropic regularized version to minimize:

$$\mathcal{W}_{c,\boldsymbol{\pi}}^\epsilon\left(\mathbb{Q}^T, \mathcal{Q}^S\right) = \min_A \left\{ \sum_{i=1}^{N_T} \sum_{m=1}^{M} a_{im} c\left(G\left(\mathbf{x}_i\right), \mathbb{Q}_m^S\right) \right.$$
$$\left. -\epsilon H(A) : \sum_{m=1}^{M} a_{im} = \frac{1}{N_T}, \sum_{i=1}^{N_T} a_{im} = \pi_m \right\}, \quad (8)$$

where $H(A) \coloneqq -\sum_{i=1}^{N_T} \sum_{m=1}^{M} a_{im} \log a_{im}$ denotes an entropy of the transportation matrix $A$, and $\epsilon$ is the regularization rate. During the training, we use Sinkhorn algorithm [Cuturi, 2013] to solve this OP and achieve $A^*$ at every mini-batch. Interestingly, the solution of Eq. (8) also provides us $\sum_{m=1}^{M} a_{im}^* = \frac{1}{N_T}$ or in other words, $N_T \sum_{m=1}^{M} a_{im}^* = 1$. Hence, we can define the pseudo label $\hat{y}_i^T \coloneqq N_T a_i^*$ for a given target sample $\mathbf{x}_i$ and it satisfies $\sum_{m=1}^{M} \hat{y}_{im}^T = N_T \sum_{m=1}^{M} a_{im}^* = 1$. The definition of $\hat{y}_i^T$ is then used for minimizing $\mathcal{L}^{pl}$ in Eq. (7).

One problem when choosing $\hat{y}_i^T \coloneqq N_T a_i^*$ is that the performance of the teacher $h^T$ can be reduced if some pseudo

labels are incorrect, especially at the beginning of the training due to the data and label shifts between the source and target domains. This issue also influences the distillation process since we aim to build a well-classified teacher $h^T$ on the target domain to transfer some of its aspects (e.g, its "dark knowledge") to the student $h^S$. To avoid this problem, inspired by Yang et al. [2021], we propose only selecting highly confident pseudo labels (i.e., pseudo labels whose entropies are less than a threshold) using an entropy-based selection method. The OP in Eq. (7) is now minimized w.r.t. the weights $w_i$:

$$\mathcal{L}_w^{pl} = \frac{1}{n_T} \sum_{i=1}^{n_T} w_i CE\left(\sigma\left(h^T\left(G\left(\mathbf{x}_i\right)\right)\right), \hat{y}_i^T\right), \quad (9)$$

where $w_i = \mathbb{I}_{\left\{H\left(\hat{y}_i^T\right) < H_\rho\right\}}$ with $\mathbb{I}_C$ representing the indicator function for a statement $C$ (i.e., $\mathbb{I}_C$ returns 1 iff $C$ is true), $H\left(\hat{y}_i^T\right) \coloneqq -\sum_{m=1}^{M} a_{im} \log a_{im}$ is the entropy of a pseudo label $\hat{y}_i^T$ w.r.t. a target example $\mathbf{x}_i$, and the threshold $H_\rho$ denotes the $\rho$-th percentile of $H\left(\hat{y}_i^T\right)$.

Additionally, when training our COOK, at each iteration, we sample a mini-batch of target examples and consider $\mathbb{Q}^T$ as the distribution of latent representations corresponding to this mini-batch. Therefore, $N_T$ in Eq. (8) is replaced by the batch size and the threshold $H_\rho$ denotes the $\rho$-th percentile of $H\left(\hat{y}_i^T\right)$ in the mini-batch.

Finally, we present our framework in Figure 2 which includes three main steps: (i) the teacher is encouraged to be an expert on the target domain using the pseudo labeling technique; (ii) the teacher transfers its knowledge to the student via a distillation process to support the student to generalize well on the target domain; and (iii) the predicted probabilities of the student classifier are utilized for minimizing $\mathcal{W}_{c,\boldsymbol{\pi}}^\epsilon\left(\mathbb{Q}^T, \mathcal{Q}^S\right)$ using Sinkhorn algorithm, and offering the optimal transportation matrix $A^*$ to compute pseudo labels. The pseudo labels with low entropies are selected to train the teacher at the first step. This process forms a closed cycle in which target samples are confidently moved towards corresponding source class-conditional distributions $\mathbb{Q}_m^S$ under the consistently cyclic guidance of the key factors including the distributional optimal transport and knowledge distillation, which motivates us to propose our COOK.

## 5.3 TRAINING PROCEDURE OF COOK

To strengthen $h^S$ for providing better predictions and accelerating matching target samples $\mathbf{x}^T$ to source class-conditional distributions $\mathbb{Q}_m^S$, we enforce the clustering assumption to $h^S$. Inspired by applying clustering assumption in domain adaptation works [Shu et al., 2018, Kumar et al., 2018], we employ Virtual Adversarial Training (VAT) [Miyato et al., 2019] in conjunction with minimizing entropy [Grandvalet and Bengio, 2005] of the prediction of

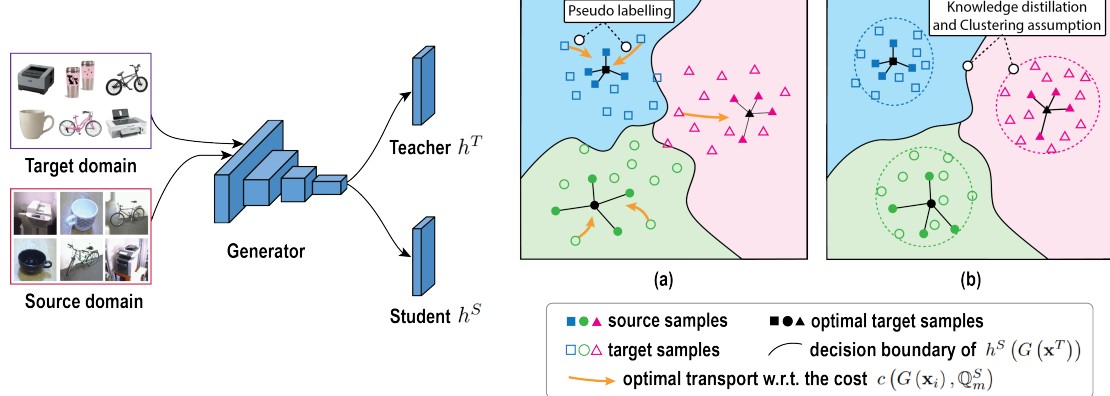

Figure 3: The overall architecture of our proposed method where $G$ is a weight-sharing generator for mapping the source and target data into the latent space. The teacher $h^T$ and the student $h^S$ act in a cyclic process as described in Figure 2 where we apply pseudo labelling, knowledge distillation and enforce clustering assumption: (a) when minimizing pseudo labelling loss $\mathcal{L}_w^{pl}$, target samples are encouraged to move towards the corresponding source class-conditional distributions; (b) minimizing distillation loss $\mathcal{L}^{dl}$ pushes the target samples closer to the source samples due to the distillation process between predictions of the teacher and student classifiers. While minimizing $\mathcal{L}^{clus}$ accelerates transporting target samples, achieves a strong clustering, improves local smoothness and achieves the good generalization ability of $h^S$ on the target domain, from which the pseudo labels are selected with the high confidence.

$h^S\left(G\left(\mathbf{x}^T\right)\right)$. VAT is an effective technique to improve the local distribution robustness [Nguyen-Duc et al., 2022, Phan et al., 2022]. At first, given a target sample $\mathbf{x}$, a perturbation of $\mathbf{x}$, which is $\mathbf{x}'$ that makes the student classifier $h^S$ give a different prediction from $\mathbf{x}$ is chosen. And then $h^S$ is enforced to predict the same label for $\mathbf{x}$ and $\mathbf{x}'$. As a result, the decision boundary of $h^S$ is pushed away from the target sample $\mathbf{x}$, which achieves a better generalization ability for $h^S$ on the target domain.

$$\mathcal{L}^{clus} = \mathcal{L}^{ent} + \mathcal{L}^{vat}, \qquad (10)$$

where with $H$ to be the entropy, we have defined:

$\mathcal{L}^{ent} = \mathbb{E}_{\mathbb{P}^T}\left[H\left(\sigma\left(h^S\left(G\left(\mathbf{x}\right)\right)\right)\right)\right],$

$\mathcal{L}^{vat} = \mathbb{E}_{\mathbb{P}^T}\left[\max_{\mathbf{x}':\|\mathbf{x}'-\mathbf{x}\|<\theta} D_{KL}\Big(\sigma\left(h^S\left(G\left(\mathbf{x}\right)\right)\right),\right.$

$\left.\sigma\left(h^S\left(G\left(\mathbf{x}'\right)\right)\right)\Big)\right]$, where $D_{KL}$ denotes the Kullback-Leibler divergence and $\theta$ is a hyperparameter set to a very small positive number.

The final optimization problem of our COOK for finding $h^S, h^T$ and $G$ is as follows:

$$\min_{h^S, h^T, G}\left\{\mathcal{L}^{src} + \alpha\mathcal{L}^{dl} + \beta\mathcal{L}_w^{pl} + \gamma\mathcal{L}^{clus}\right\}, \qquad (11)$$

where $\alpha, \beta, \gamma > 0$ are trade-off parameters. Conveniently, the cyclic process in Figure 2 is operated synchronously by simultaneously updating $h^S, h^T$ and $G$ during the training. Finally, we present the key steps of our COOK in Algorithm

1 and the overall architecture and motivation of component losses are depicted in Figure 3.

---

**Algorithm 1** Pseudocode for training our proposed COOK.

**Input:** A source batch $\mathcal{B}^S = \left\{\left(\mathbf{x}_i^S, y_i^S\right)\right\}_{i=1}^b$, a target batch $\mathcal{B}^T = \left\{\mathbf{x}_j^T\right\}_{j=1}^b$ ($b$ denotes the batch size).

**Output:** Classifiers $h^{S*}, h^{T*}$, generator $G^*$.

1: **for** number of training iterations **do**
2:     Solve the OP in Eq. (8) using Sinkhorn algorithm to find $A^*$.
3:     Compute $\hat{y}_i^T$ in Eq. (9) based on $A^*$.
4:     Compute $w_i$ in Eq. (9) based on $H_\rho$.
5:     Update $h^S, h^T$ and $G$ according to Eq. (11).
6: **end for**

---

## 6 EXPERIMENTS

In this section, we conduct experiments on benchmark datasets including *Office-31*, *Office-Home*, and *ImageCLEF-DA* to compare with existing baselines, especially OT-based and class-aware UDA methods.

### 6.1 DATASETS

**Office-31** [Saenko et al., 2010] is a well-known public dataset used for UDA. It consists of three domains including Amazon (**A**), Webcam (**W**) and Dslr (**D**) with 31 common classes and 4,110 images in total.

**Office-Home** [Venkateswara et al., 2017] is another and

more challenging dataset for UDA which contains images from four different domains, namely Artistic (**Ar**), Clip Art (**Cl**), Product (**Pr**) and Real-world images (**Re**). This dataset consists of around 15,588 images in total with 65 object categories in office and home scenes.

**ImageCLEF-DA** [Caputo et al., 2014] includes three domains including Caltech-256 (**C**), ImageNet ILSVRC 2012 (**I**), and Pascal VOC 2012 (**P**), each of which has 12 classes with 50 images per class.

## 6.2 IMPLEMENTATION DETAILS

In the experiments on the *Office-31*, *Office-Home* and *ImageCLEF-DA* datasets, we use the extracted features (2048 dimensions) from ResNet-50 [He et al., 2016]. The generator includes a fully connected layer that outputs 256 dimensions. We use the same architecture for the student and teacher networks which consists of a fully connected layer for each network.

Some hyperparameters substantially contributes to model performance, namely the temperature $\tau$ in Eq. (6), and the percentile $\rho$ in Eq. (9). As suggested in the ablation study, we choose $\tau = 10.0$ to effectively activate the knowledge distillation process from the teacher to the student. The percentile $\rho$ is important to measure how well the student $h^S$ can generalize on the target domain. We empirically find that $\rho = 20$ or in other words, choosing the 20-th percentile of $H\left(\hat{y}_i^T\right)$ is appropriate to select high-confidence pseudo labels. Additionally, setting $\epsilon$ less than or equal to $0.1$ can achieve better performance and we set $\epsilon$ to $0.1$. We also select the trade-off parameters $\alpha = \beta = 1.0$ and $\gamma = 0.1$ in our experiments as suggested in the ablation studies.

We apply Adam optimizer [Kingma and Ba, 2015] ($\beta_1 = 0.5, \beta_2 = 0.999$) with Polyak averaging [Polyak and Juditsky, 1992], and the learning rate is set to $10^{-4}$ for *Office-31* and *Office-Home*, and $5 \times 10^{-5}$ for *ImageCLEF-DA*. For the baselines, we report the experimental results mentioned in the original papers. It is noticeable that in all experiments, we only train the feature extractor, and the performance of COOK can be further improved when fine-tuning the backbone ResNet-50 is conducted.

## 6.3 RESULT AND DISCUSSION

We compare our COOK with the standard baseline ResNet-50 [He et al., 2016] and existing works including DAN [Long et al., 2015], DANN [Ganin and Lempitsky, 2015], RTN [Long et al., 2016], iCAN [Zhang et al., 2018], CDAN-E [Long et al., 2018], CDAN-BSP [Chen et al., 2019], CDAN-T [Wang et al., 2019a], TPN [Pan et al., 2019], rRevGrad+CAT [Deng et al., 2019], CADA-P [Kurmi et al., 2019], SymNets [Zhang et al., 2019], especially class-aware DA and OT-based methods, namely RADA [Wang et al.,

Table 1: Mean accuracy (%) on Office-31 for unsupervised domain adaptation (ResNet-50).

| Method | A→W | A→D | D→W | W→D | D→A | W→A | Avg |
|---|---|---|---|---|---|---|---|
| ResNet-50 | 68.4 | 68.9 | 96.7 | 99.3 | 62.5 | 60.7 | 76.1 |
| DAN | 80.5 | 78.6 | 97.1 | 99.6 | 63.6 | 62.8 | 80.4 |
| DANN | 82.0 | 79.7 | 96.9 | 99.1 | 68.2 | 67.4 | 82.2 |
| RTN | 84.5 | 77.5 | 96.8 | 99.4 | 66.2 | 64.8 | 81.6 |
| iCAN | 92.5 | 90.1 | 98.8 | **100.0** | 72.1 | 69.9 | 87.2 |
| CDAN-E | 94.1 | 92.9 | 98.6 | **100.0** | 71.0 | 69.3 | 87.7 |
| CDAN-BSP | 93.3 | 93.0 | 98.2 | **100.0** | 73.6 | 72.6 | 88.5 |
| CDAN-T | 95.7 | 94.0 | 98.7 | **100.0** | 73.4 | 74.2 | 89.3 |
| TPN | 91.2 | 89.9 | 97.7 | 99.5 | 70.5 | 73.5 | 87.1 |
| rRevGrad+CAT | 94.4 | 90.8 | 98.0 | **100.0** | 72.2 | 70.2 | 87.6 |
| SymNets | 90.8 | 93.9 | 98.8 | **100.0** | 74.6 | 72.5 | 88.4 |
| DeepJDOT | 88.9 | 88.2 | 98.5 | 99.6 | 72.1 | 70.1 | 86.2 |
| ETD | 92.1 | 88.0 | 100.0 | **100.0** | 71.0 | 69.3 | 86.2 |
| RWOT | **95.1** | 94.5 | 99.5 | **100.0** | 77.5 | 77.9 | 90.8 |
| RADA | 91.5 | 90.7 | 98.9 | **100.0** | 71.5 | 71.3 | 87.3 |
| CAN | 94.5 | 95.0 | 99.1 | 99.8 | 78.0 | 77.0 | 90.6 |
| **COOK** | **95.1** | **96.2** | 98.3 | 99.9 | **88.7** | **86.2** | **94.1** |

2019b], CAN [Kang et al., 2019], DeepJDOT [Damodaran et al., 2018], ETD [Li et al., 2020], and RWOT [Xu et al., 2020].

The results trained on *Office-31* are reported in Table 1. In general, our proposed method achieves high results with four transfer tasks greater than $95\%$. Except for the transfer tasks **D→W** and **W→D**, our model significantly outperforms others on almost adaptation tasks, and obtain $94.1\%$ on average, which is a $3.5\%$ increase compared to the runner-up result. It is worth noting that our COOK outperforms the baselines by a large margin on challenging tasks, e.g., a $10.7\%$ increase on **D→A** and **W→A** with a $9.2\%$ improvement, in which the background of the training images between the two domains are totally dissimilar.

We present the results trained on *Office-Home* in Table 2. In this dataset, our COOK surpasses 7 over 12 transfer tasks compared with the baselines and achieves the best performance, making a $2.8\%$ improvement on average. More specifically, our model sees a remarkable improvement on more challenging adaptation tasks, namely **Ar→Pr** ($3.6\%$), **Cl→Pr** ($7.6\%$), **Cl→Re** ($4.1\%$).

We further evaluate our COOK on *ImageCLEF-DA* and report the classification accuracy in Table 3. Our COOK outperforms 4 over 6 transfer tasks with an average accuracy of $90.7\%$, compared to ETD and RWOT with $89.7\%$ and $90.3\%$, respectively.

## 6.4 ANALYSIS

### 6.4.1 Hyperparameter Sensitivity and Quantitative Evaluation

We conduct experiments to evaluate hyperparameter sensitivity and quantitative result for our proposed COOK in Figure 4. Figure 4a experiences a decrease of the model per-

Table 2: Mean accuracy (%) on Office-Home for unsupervised domain adaptation (ResNet-50).

| Method | Ar→Cl | Ar→Pr | Ar→Re | Cl→Ar | Cl→Pr | Cl→Re | Pr→Ar | Pr→Cl | Pr→Re | Re→Ar | Re→Cl | Re→Pr | Avg |
|---|---|---|---|---|---|---|---|---|---|---|---|---|---|
| ResNet-50 | 34.9 | 50.0 | 58.0 | 37.4 | 41.9 | 46.2 | 38.5 | 31.2 | 60.4 | 53.9 | 41.2 | 59.9 | 46.1 |
| DAN | 43.6 | 57.0 | 67.9 | 45.8 | 56.5 | 60.4 | 44.0 | 43.6 | 67.7 | 63.1 | 51.5 | 74.3 | 56.3 |
| DANN | 45.6 | 59.3 | 70.1 | 47.0 | 58.5 | 60.9 | 46.1 | 43.7 | 68.5 | 63.2 | 51.8 | 76.8 | 57.6 |
| SymNets | 47.7 | 72.9 | 78.5 | 64.2 | 71.3 | 74.2 | 64.2 | 48.8 | 79.5 | **74.5** | 52.6 | 82.7 | 67.6 |
| CDAN-E | 50.7 | 70.6 | 76.0 | 57.6 | 70.0 | 70.0 | 57.4 | 50.9 | 77.3 | 70.9 | 56.7 | 81.6 | 65.8 |
| CDAN-BSP | 52.0 | 68.6 | 76.1 | 58.0 | 70.3 | 70.2 | 58.6 | 50.2 | 77.6 | 72.2 | **59.3** | 81.9 | 66.3 |
| CDAN-T | 50.2 | 71.4 | 77.4 | 59.3 | 72.7 | 73.1 | 61.0 | **53.1** | 79.5 | 71.9 | 59.0 | 82.9 | 67.6 |
| DeepJDOT | 48.2 | 69.2 | 74.5 | 58.5 | 69.1 | 71.1 | 56.3 | 46.0 | 76.5 | 68.0 | 52.7 | 80.9 | 64.3 |
| ETD | 51.3 | 71.9 | **85.7** | 57.6 | 69.2 | 73.7 | 57.8 | 51.2 | 79.3 | 70.2 | 57.5 | 82.1 | 67.3 |
| RWOT | **55.2** | 72.5 | 78.0 | 63.5 | 72.5 | 75.1 | 60.2 | 48.5 | 78.9 | 69.8 | 54.8 | 82.5 | 67.6 |
| **COOK** | 53.0 | **76.5** | 81.8 | **65.5** | **80.3** | **79.2** | **64.5** | 51.8 | **82.4** | 71.3 | 54.2 | **83.9** | **70.4** |

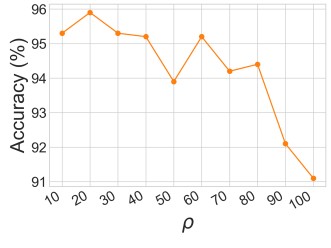

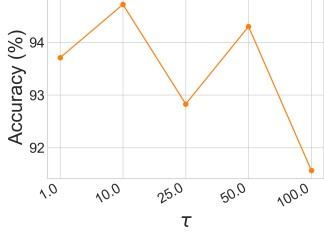

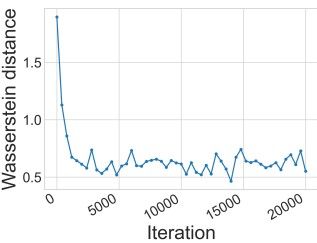

(a) The changes of $\rho$-percentile.

(b) Study of twisting $\tau$.

(c) Values of $\mathcal{W}^{\epsilon}_{c,\pi}\left(\mathbb{Q}^T, \mathcal{Q}^S\right)$.

Figure 4: Ablation studies of our proposed method on the transfer task **A→W**.

Table 3: Mean accuracy (%) on ImageCLEF-DA for unsupervised domain adaptation (ResNet-50).

| Method | I→P | P→I | I→C | C→I | C→P | P→C | Avg |
|---|---|---|---|---|---|---|---|
| RTN | 75.6 | 86.8 | 95.3 | 86.9 | 72.7 | 92.2 | 84.9 |
| iCAN | 79.5 | 89.7 | 94.7 | 89.9 | 78.5 | 92.0 | 87.4 |
| CDAN-E | 77.7 | 90.7 | 97.7 | 91.3 | 74.2 | 94.3 | 87.7 |
| CDAN-T | 78.3 | 90.8 | 96.7 | 92.3 | 78.0 | 94.8 | 88.5 |
| SymNets | 80.2 | 93.6 | 97.0 | 93.4 | 78.7 | 96.4 | 89.9 |
| CADA-P | 78.0 | 90.5 | 96.7 | 92.0 | 77.2 | 95.5 | 88.3 |
| DeepJDOT | 77.7 | 90.6 | 95.1 | 88.5 | 75.3 | 94.3 | 86.9 |
| ETD | 81.0 | 91.7 | 97.9 | 93.3 | 79.5 | 95.0 | 89.7 |
| RWOT | **81.5** | 93.1 | **98.0** | 92.8 | 79.3 | 96.8 | 90.3 |
| RADA | 79.2 | 92.4 | 97.5 | 91.1 | 76.6 | 95.3 | 88.7 |
| **COOK** | 80.1 | **95.5** | 97.0 | **95.9** | 79.1 | 96.3 | **90.7** |

Table 4: Accuracy (%) of ablation study on ImageCLEF-DA.

| $\mathcal{L}^{src}$ | $\mathcal{L}^{pl}_w$ | $\mathcal{L}^{dl}$ | $\mathcal{L}^{clus}$ | I→P | P→I | I→C | C→I | C→P | P→C | Avg |
|---|---|---|---|---|---|---|---|---|---|---|
| ✓ | ✓ | | | 75.9 | 86.1 | 93.9 | 89.0 | 74.4 | 87.4 | 84.5 |
| ✓ | ✓ | ✓ | | 76.4 | 86.6 | 93.9 | 89.9 | 76.0 | 91.2 | 85.7 |
| ✓ | ✓ | | ✓ | 78.6 | 91.0 | 95.9 | 92.9 | 77.9 | 95.9 | 88.7 |
| ✓ | | ✓ | ✓ | 78.5 | 90.9 | 96.7 | 93.3 | **79.6** | 95.0 | 89.0 |
| ✓ | ✓ | ✓ | ✓ | **80.1** | **95.5** | **97.0** | **95.9** | 79.1 | **96.3** | **90.7** |

Table 5: Results (%) on different training strategies.

| Methods | A→W | A→D | D→W | W→D | D→A | W→A | Avg |
|---|---|---|---|---|---|---|---|
| Without KD | 93.8 | 94.6 | 97.8 | 99.2 | 86.4 | 86.1 | 93.0 |
| With KD | **95.1** | **96.2** | **98.3** | **99.9** | **88.7** | **86.2** | **94.1** |

formance when twisting $\rho$ in Eq. (9). Our proposed COOK works well with $\rho$ from 10 to 40. Relying on this investigation, we pick $\rho = 20$ in our experiments. Similarly, Figure 4a shows results with the changes of $\tau$. We search $\tau$ in the grid of $\{1.0, 10.0, 25.0, 50.0, 100.0\}$ and find that setting $\tau = 10.0$ achieves the best performance to perform knowledge distillation. Furthermore, we investigate the Wasserstein distance $\mathcal{W}^{\epsilon}_{c,\pi}\left(\mathbb{Q}^T, \mathcal{Q}^S\right)$ in Figure 4c, which sees a reduction during the training. This result shows the success of transporting target samples to their corresponding source class-conditional distributions.

We further evaluate the effects of the trade-off parameters $\alpha, \beta, \gamma$ on model performance by twisting their values. Figure 5 shows results when we search $\alpha, \beta$ and $\gamma$ in the grid of $\{0.001, 0.01, 0.1, 1.0, 5.0, 10.0\}$ and report the test accuracy on two transfer tasks $\mathbf{P} \rightarrow \mathbf{I}$ (*ImageCLEF-DA*) and $\mathbf{A} \rightarrow \mathbf{D}$ (*Office-31*). The results show that the model yields the stable performance when $\alpha, \beta, \gamma$ from 0.001 to 1.0. We find that our COOK can achieve high performance when $\alpha = \beta = 1.0$ and $\gamma = 0.1$, hence we suggest picking these values on most of our experiments.

#### 6.4.2 Effect of Losses

We investigate the effectiveness of the pseudo labelling loss $\mathcal{L}^{pl}_w$, the distillation loss $\mathcal{L}^{dl}$, and the clustering assumption loss $\mathcal{L}^{clus}$ in Eq. (11). The experimental results are described in Table 4, which shows that all component losses contribute to the model performance since they participate

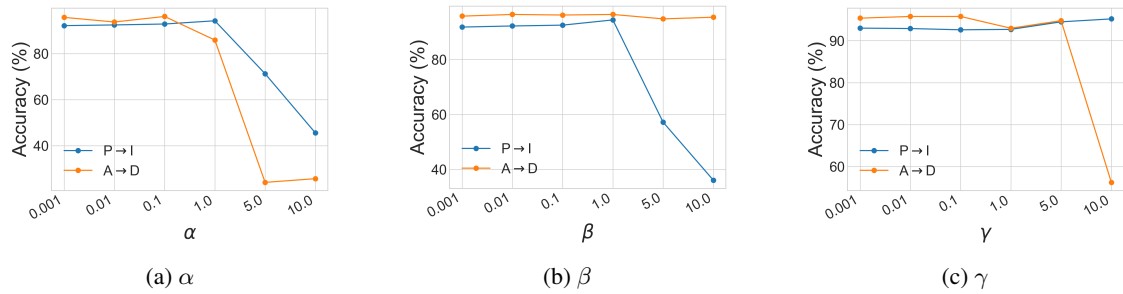

|(a) $\alpha$|(b) $\beta$|(c) $\gamma$|

Figure 5: Analysis of hyperparameter sensitivity of $\alpha, \beta$ and $\gamma$ on transfer tasks $\mathbf{P} \rightarrow \mathbf{I}$ and $\mathbf{A} \rightarrow \mathbf{D}$.

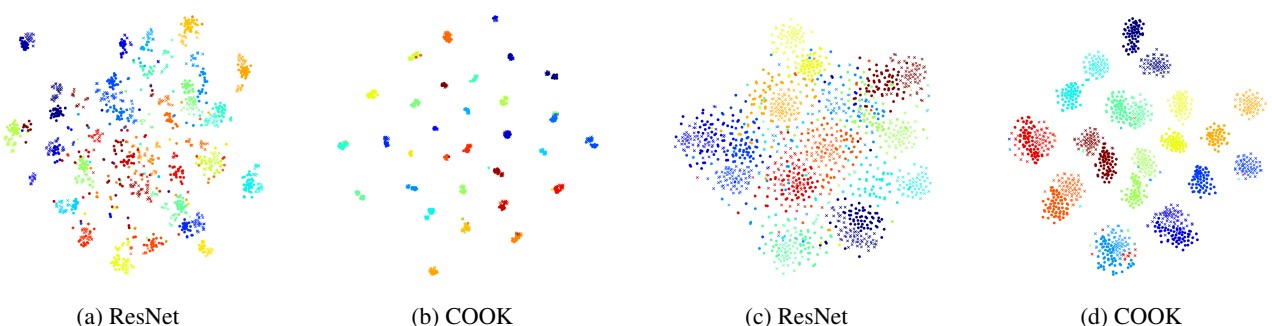

(a) ResNet        (b) COOK        (c) ResNet        (d) COOK

Figure 6: The t-SNE visualization of $\mathbf{A} \rightarrow \mathbf{W}$ (Figure a, b) and $\mathbf{P} \rightarrow \mathbf{C}$ (Figure c, d) tasks with label and domain information. Each color denotes a class while the circle and cross markers represent the source and target data respectively.

in the cyclic process and support to match target samples to the corresponding source regions. It is noticeable that the model performance is the best when all component losses are activated and participate in the training process.

### 6.4.3 Effect of Knowledge Distillation

We further testify the contribution of KD to our proposed method in two different scenarios: *Without KD* and *With KD*. For *Without KD* setting, we deploy a model where $h^S$ and $h^T$ are weight-sharing networks and train this model using the final optimization problem where $\mathcal{L}^{dl} = 0$. We compare the *Without KD* setting with our architecture COOK (a.k.a. *With KD*) and report the accuracy score in Table 5. The results show that our COOK with KD outperforms that without KD by nearly $1\%$, which demonstrates the effectiveness of KD for our framework.

### 6.4.4 Feature Visualization

We select transfer tasks $\mathbf{A} \rightarrow \mathbf{W}$ (*Office-31*) and $\mathbf{P} \rightarrow \mathbf{C}$ (*ImageCLEF-DA*) tasks to visualize their representation in the latent space using *t*-SNE [van der Maaten and Hinton, 2008]. The visualizations in Figure 6a and 6c show that after going through the backbone model ResNet-50, there is still a mismatch between the source and target distributions due to the data and label shifts. However, our proposed COOK

(see Figure 6b and 6d) is trained to transport target samples to source samples, which closes this gap and achieves better alignment between the target and the source samples.

## 7 CONCLUSION

In this paper, we develop a novel framework for class-aware unsupervised domain adaptation. In particular, our proposed method is based on the proposed distributional OT which quantifies an OT distance between a distribution of target data and the source class-conditional distributions. To efficiently train our model with the proposed distributional OT, we develop a novel model operating in a cyclic process. By incorporating knowledge distillation and pseudo labelling technique into this process, our proposed COOK effectively tackles the data and label shifts problem by transporting the target samples to the corresponding source class-conditional distributions in a class-aware manner. The experimental results show that COOK outperforms existing UDA methods, especially the class-aware and OT-based ones on the benchmark datasets.

**Acknowledgements**

This work was supported by the US Air Force grant FA2386-21-1-4049.

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
