# OpenReview forum: "Cycle Class Consistency with Distributional Optimal Transport and Knowledge Distillation for Unsupervised Domain Adaptation"
_auai.org/UAI/2022/Conference — UAI 2022 Poster_

### Official Review · Reviewer_rTSN · 2022-04-12

**Q2(1) Originality/Novelty:** 3
**Q2(2) Significance/Impact:** 3
**Q2(3) Correctness/Technical Quality:** 3
**Q2(6) Clarity Of Writing:** 3
**Q6 Overall Score:** 7
**Q8 Confidence In Your Score:** 4

**Q1 Summary And Contributions:**

The paper proposes a new domain adaptation method, based on optimal transport between target samples and source classes, combined with a distillation between source and target classifiers. Experiments are performed that show an improvement upon the previous state-of-the-art.

**Q2 Assessment Of The Paper:**

More detailed information regarding each of these aspects is given below:

**Q2(4) Quality Of Experiments (Optional):**

2: Fair: The experimental evaluation is weak: important baselines are missing, or the results do not adequately support the main claims.

**Q2(5) Reproducibility:**

3: Good: Key resources (e.g., proofs, code, data) are available and key details (e.g., proofs, experimental setup) are sufficiently well-described for competent researchers to confidently reproduce the main results.

**Q3 Main Strengths:**

* Very good results
* An ablation study is performed to show that all parts of the model are important, and sensitivity to hyperparameters.

**Q4 Main Weakness:**

* The final method is very complicated, with five different terms in the loss function, many hyperparameters, and multiple classifier models.

* The intuition behind the method is not explained well.

* It is not clear how the ablation studies were carried out. What datasets were used? How do we know that the reported results are not simply the best out of many runs with different parameter values? Was a validation set used?

* Some details are not clear from the paper, such as "we employ Virtual Adversarial Training (VAT)", which involves selecting adversarial perturbations. This is completely glossed over.


**Q5 Detailed Comments To The Authors:**

* In the model distillation, why is h^t seen as the teacher? It works nicely with T standing for both teacher and target, but intuitively, I would expect that you are able to train a model on the source domain from which another model can learn. How can you learn a student model from a bad teacher trained only with pseudo labels.

* What is the optimal transport problem used for exactly? The labels are inferred with pseudo-labeling, what remains learnable in (3) or (8)?

* Standard deviation of the results over multiple runs should be included in the tables. This is common in other DA papers.

* The transport cost c(x, P^S_m) is defined as the class conditional likelihood p(x | y=m), but to compute this you would need a density estimator, not just a classifier. A classifier will give the converse, p(y=m | x)

* The OT problem in (8) enforces a certain class balance, as specified by π. How is π chosen? Are these determined empirically from the source domain?

* Using h^T for teacher and h^S for student is confusing when S and T superscripts are also used for source and target domain.

* The name Cycle Consistent makes me think of cycle consistency loss in CycleGANs, but that is completely unrelated to this work.


**Q7 Justification For Your Score:**

The results are good, the method is explained in sufficient detail.
However, the method is complicated and could be motivated better.
I also have some concerns about the hyperparameter selection.

**Q9 Complying With Reviewing Instructions:**

0: No.

---

### Official Review · Reviewer_1mtQ · 2022-04-13

**Q2(1) Originality/Novelty:** 3
**Q2(2) Significance/Impact:** 2
**Q2(3) Correctness/Technical Quality:** 3
**Q2(6) Clarity Of Writing:** 3
**Q6 Overall Score:** 4
**Q8 Confidence In Your Score:** 3

**Q1 Summary And Contributions:**

The paper tackles unsupervised domain adaptation (UDA) via a cycle class-consistent model combining optimal transport (OT) and knowledge distillation (KD). To achieve computation efficiency, they regularize the OT problem with negative entropy to obtain a pseudo labels, i.e. solving the regularized/"soft" problem instead of the original one, to learn the teacher. Knowledge distillation is then utilized to distill and transfer the teacher's knowledge to the student.

**Q2 Assessment Of The Paper:**

More detailed information regarding each of these aspects is given below:

**Q2(4) Quality Of Experiments (Optional):**

3: Good: The experimental evaluation is adequate, and the results convincingly support the main claims.

**Q2(5) Reproducibility:**

3: Good: Key resources (e.g., proofs, code, data) are available and key details (e.g., proofs, experimental setup) are sufficiently well-described for competent researchers to confidently reproduce the main results.

**Q3 Main Strengths:**

The idea of OT for domain adaptation is not new. This work differs a bit perhaps in the way the cost function is defined, taking into account target and source class-conditional distributions. However, the combination of OT and KD, which is again a known method, is a good step to verify if such combination could work or improve over known methods. The work has methodological contribution rather than theories, so it's important to illustrate its performance in practice. In this aspects, the experiments are extensive and show marginal outperformance of the proposed method over SOTAs.

**Q4 Main Weakness:**

The experiments show mostly marginal improvement over SOTAs, which could have been an effect of fine tuning rather than the superiority of the proposed method.

**Q5 Detailed Comments To The Authors:**

Regularizing OT with negative entropy would result in dense and strictly positive solution, i.e. pseudo label, which could affect the performance. It may be worth solving the original problem without regularization to see its performance (note: the original one is simple linear optimization, so there should be empirical solvers). Another point is that using smaller $\epsilon$ than 0.1, the regularizing coefficients of entropy, may result in better performance perhaps. The negative entropy is of order $-H(X) =O( \log(n))$, so usually $\epsilon$ is set to be ~ $\epsilon = O(desiredError / \log(n))$ which can be small. My point is that a bit more tuning of $\epsilon$ may help.

**Q7 Justification For Your Score:**

It is a nice method, yet the empirical results are not strong enough. Given that the nature of this work is not theoretical, I believe more favorable performance is desired.

**Q9 Complying With Reviewing Instructions:**

1: Yes.

---

### Official Review · Reviewer_XPWv · 2022-04-18

**Q2(1) Originality/Novelty:** 2
**Q2(2) Significance/Impact:** 2
**Q2(3) Correctness/Technical Quality:** 2
**Q2(6) Clarity Of Writing:** 3
**Q6 Overall Score:** 5
**Q8 Confidence In Your Score:** 4

**Q1 Summary And Contributions:**

This paper presents an unsupervised class aware domain adaptation approach. The approach is based on distributional optimal transport and seeks to align unlabeled target examples distribution with labeled source class-conditioned distributions. The model is trained in a cyclic manner involving pseudo-labeling, knowledge distillation, and minimizing the optimal distribution transport cost. Experimental results suggest performance improvement of proposed approach on three benchmark datasets.

**Q2 Assessment Of The Paper:**

More detailed information regarding each of these aspects is given below:

**Q2(4) Quality Of Experiments (Optional):**

3: Good: The experimental evaluation is adequate, and the results convincingly support the main claims.

**Q2(5) Reproducibility:**

3: Good: Key resources (e.g., proofs, code, data) are available and key details (e.g., proofs, experimental setup) are sufficiently well-described for competent researchers to confidently reproduce the main results.

**Q3 Main Strengths:**

- Paper is well-written and the presentation is good

- Combines pseudo-labeling with student-teacher and optimal transport of distributions


**Q4 Main Weakness:**

- Theoretical understanding of the proposed approach would further improve the quality of the paper. It is not clear if entropy regularized approach has any particular limitations

- Some implementation details of how the losses are optimized could be useful.


**Q5 Detailed Comments To The Authors:**

- It would be useful to comment on semi-supervised domain adaptation and class-aware unsupervised domain adaptation. As such, the class aware unsupervised domain adaptation is highly sensitive to the pseudo-labeling strategy. In challenging problems, it is likely that some target samples are required to achieve effective class alignment. (see [1], [2] )

- Distribution vs fixed optimal transport differences are not well explained, which seems to be the main contribution of the paper.

[1] A Dirt-t Approach to Unsupervised Domain Adaptation, 2018

[2] DIRL: Domain-Invariant Representation Learning for Sim-to-Real Transfer, 2020


**Q7 Justification For Your Score:**

See above. Experienced in publishing and reviewing papers in domain adaptation

**Q9 Complying With Reviewing Instructions:**

1: Yes.

---

### Decision · Program_Chairs · 2022-05-15

**Decision:**

Accept (Poster)

**Comment:**

Meta Review: This paper proposes an unsupervised class-aware domain adaptation approach, which is based on distributional optimal transport and seeks to align unlabeled target examples distribution with labeled source class-conditioned distributions. The model is trained in a cyclic manner involving pseudo-labeling, knowledge distillation, and minimizing the optimal distribution transport cost. The authors conduct experiments to demonstrate the performance improvement of proposed approach on three benchmark datasets. All the reviewers agree that the paper is well written and easy to understand. Review XPWv and 1mtQ agree that the idea of OT for domain adaptation itself is not new, but the authors explore the combination of OT and KD and improve the existing methods. R2 is the only reviewer who is against accepting this submission, and R2's only concern is about whether the small differences to baseline are significant. The authors responded by showing the significant performance on the Office-31 dataset. Due to the reasons above, I make the decision to accept the paper.